# Exploring drivers of unsafe disposal of child stool in India using hierarchical regression model

Margubur Rahaman[1]*, Avijit Roy[2], Pradip Chouhan[3]*, Md. Juel Rana[4]

1 Department of Migration & Urban Studies, International Institute for Population Sciences (IIPS), Deonar, Mumbai, India, 2 Department of Geography, Malda College, State Aided College Teacher, Malda, West Bengal, India, 3 Department of Geography, University of Gour Banga, Malda, West Bengal, India, 4 Govind Ballabh Pant Social Science Institute (GBPSSI), Allahabad, India

* margubur48@gmail.com (MR); pradipchouhanmalda@gmail.com (PC)

## Abstract

### Background

Disposal of children's stools is often neglected in Indian sanitation programs, putting them at higher risk of diseases transmitted through the fecal-oral route. Therefore, the current study aims to identify the socioeconomic and demographic factors associated with the unsafe disposal of child stool in India and to estimate the geographical variation in unsafe disposal.

### Methods

The study used 78,074 births under two years from the fifth round of the National Family Health Survey (2019–21). Descriptive statistics, bivariate analysis with the chi-square test, and a four-level hierarchical logistic regression model were applied to accomplish the study objectives.

### Results

Findings revealed a 61.3% prevalence of unsafe stool disposal nationwide, significantly varying between rural (45%) and urban (67%) areas. Multilevel logistic regression highlighted that mother's education, wealth quintile, and sanitation facility were significant predictors of unsafe disposal of child stools. Random intercept statistics revealed a substantial geographical unit-level variance in unsafe stool practice in India.

### Conclusion

The study emphasizes the widespread unsafe disposal of child stool among Indian mothers with young children below two years, and the study underscores a range of contributing factors, including education, media exposure, prosperity, water availability, and sanitation. It also accentuates the significance of the geographical variance in the unsafe disposal of child stool in India, particularly at the household level, followed by the community level. Hence, the findings underscore the importance of focused interventions, including targeted

**Data Availability Statement:** The data for this study was sourced from the fifth round of the National Family Health Survey (NFHS-5), which is publicly available based on research proposal and filling out mandatory registration form, creating

user account and receiving data use approval letter through the Demographic and Health Survey DHS website. The present study was obtained data for specific research purposes only, with authorization from DHS (accession numbers: 184169). In particular, to access the present study data, users need to register for an account on the DHS website. Once registered, users can conveniently download the dataset for analysis. To access the dataset, users can follow these steps: Register for a user account at https://dhsprogram.com/data/new-user-registration.cfm, Visit the data catalogue at https://dhsprogram.com/Data/, select the available dataset at https://dhsprogram.com/data/available-datasets.cfm, Choose the country "India" under the available datasets, navigate to the dataset titled "India: Standard DHS, 2019-21" under weblink https://dhsprogram.com/data/dataset/India_Standard-DHS_2020.cfm?flag=1, select the file IABR7EDT.ZIP for download. Comprehensive instructions for accessing the DHS data are provided on the DHS website for easy reference at https://dhsprogram.com/data/Access-Instructions.cfm.

**Funding:** The author(s) received no specific funding for this work.

**Competing interests:** The authors have declared that no competing interests exist.

household-level poverty alleviation programs, initiatives to enhance sanitation and water facilities, and community-level public health awareness programs.

## Introduction

Unsafe disposal of stool in open fields, waste receptacles, drainage systems, or through burial in soil poses notable health risks because children who come into contact with such waste are susceptible to it [1]. This unsafe disposal practice increases the risk of various diseases propagated through fecal-oral transmission [2]. The mismanagement in the disposal of child stool escalates the propensity for diarrheal infections by 23% in Kenya and 6% in India [3,4]. Similarly, unsafe disposal of child stool leads to a 35% elevation in helminth infections in Bangladesh [5]. Unsafe disposal of child stool also contributes to the proliferation of waterborne diseases [6]. Beyond its immediate adverse effects, the unsafe disposal of stool from young children amplifies the potential for long-term undernutrition, particularly stunting [7]. In parallel, inadequate management of child stool heightens the likelihood of enduring cognitive impairment among this demographic group in their later lives [8]. Therefore, proper stool management is crucial to minimizing short- and long-term health risks among children [6–9].

Previous studies predicted that good management of stool disposal can improve health outcomes significantly. Enhancing the water supply, promoting hygienic practices, and properly managing child stool might mitigate approximately 361,000 annual under-five deaths [9]. Promoting proper disposal practices for child stool could help alleviate the burden of diarrheal infections, stunting, and other health challenges among children in India [3,7,10]. Hence, dedicated research focusing on the proper management of child stool becomes imperative in the present era of Sustainable Development Goals (SDG). The SDG-6 aims to ensure access to water and sanitation for all.

Recent research reveals that over half of households with young children followed unsafe disposal of stool in lower- and middle-income countries (LMICs) [11,12]. Among LMICs, India stands as an illustrative example where safe disposal of child stool and the use of child-friendly latrines remains uncommon, despite notable strides in overall sanitation progress [13]. In particular, only 36% of Indian households with young children adhere to safe disposal practices of child stool, encompassing the appropriate use of a latrine for their child waste [14]. While India may be approaching victory in its fight against unimproved sanitation practices [10], the endeavor to establish effective stool disposal management is proving to be a cautious progression.

Numerous studies have delved into the spatial disparities in the unsafe disposal of child stool, identifying factors influencing this issue and outlining its multifaceted negative impact on children's health and well-being [1,8,9,12,13,15]. These investigations primarily centered on sub-Saharan African countries, Cambodia, and Bangladesh, using the data from the Demographic Health Survey (DHS) [12–15]. Research has underscored the significance of factors such as caregiver's educational level, religious affiliation, exposure to mass media, household wealth quintile, place of residence, household sanitation facilities and practices, and regional disparities as pivotal in influencing unsafe disposal of child stool [11–13,16]. Few studies have also illuminated the qualitative dimensions, including behaviors, attitudes, and awareness, as key determinants of child stool disposal management [16]. However, investigations focused on disposal practices of child stool in the Indian context are limited [1,11,17,18], particularly employing recent large-scale sample surveys. Many studies addressing these issues have often relied on small-area primary surveys [1,17,18], failing to present a comprehensive overview of

disposal practices of child stool in India. Hence, the present study aims to understand the stool disposal practices in India. Specifically, it seeks to address several critical questions. Firstly, does the prevalence of unsafe disposal of child stool vary by the socioeconomic and demographic characteristics of mothers and households in India? Secondly, what are the significant determinants of unsafe disposal practices for child stool in India, and how do these determinants vary across different geographical levels (state, district, and cluster) in India? The current study employs comprehensive, robust statistical methods to answer the above research questions. Proportional distribution of unsafe disposal of child stool with a 95% confidence interval and standard errors [10] and multilevel logistic regression models [15,19,20] were applied to produce the results. The findings from the study will be valuable resources for policymakers, providing insights into the safe disposal of child stool in India and facilitating evidence-based policy formulation.

## Methods

### Data source and participants

In NFHS-5 (2019–21), a total of 653,144 occupied households were selected for the sample, of which 636,699 were successfully interviewed, with a 98% response rate [14]. A total of 747,176 eligible women between the ages of 15 and 49 years were identified in the interviewed households, and 724,115 of them were successfully interviewed with a 97% response rate. Overall, 111,179 eligible men aged 15–54 in households were selected for the state module, of which 101,839 men interviews were completed with a 92% response rate.

### Study sample

The present study only includes 78,074 recent birth children below 2 years. A detailed description of the sample selection procedure is presented in Fig 1. The study includes children aged 2 and below since they depend entirely on their caregivers [12–14]. The latest NFHS report has also estimated the prevalence of unsafe stools using a sample of mothers with children aged 2 and below [14]. Therefore, the study followed NFHS sample selection approaches to maintain consistency with the national report.

### Ethical statement

The NFHS-5 received ethical approval from the ethics review board of the IIPS in Mumbai, India. Additionally, the ICF International Review Board (IRB) reviewed and approved the surveys. Prior to participating in the survey, informed written consent was obtained from all respondents. Each individual's approval was obtained before conducting the interview.

### Outcome variable

The outcome variable was the disposal of the child's stool. This variable exhibits a dichotomous nature, with two distinct categories: safe disposal (coded as 0) and unsafe disposal (coded as 1). Safe disposal is characterized by appropriate management of child stool. This encompasses situations where mothers use designated toilets or latrines for stool disposal. Furthermore, fecal matter is considered appropriately handled if it is placed or rinsed into a toilet or latrine or if it is properly buried. These practices collectively define safe disposal methods [14]. Conversely, unsafe disposal of child school encompasses leaving stool in open areas, discarding it in garbage bins, rinsing it into drains, or other divergent methods. These classifications align with India's latest Demographic and Health Survey report (2022) by the International Institute for Population Sciences and ICF [14].

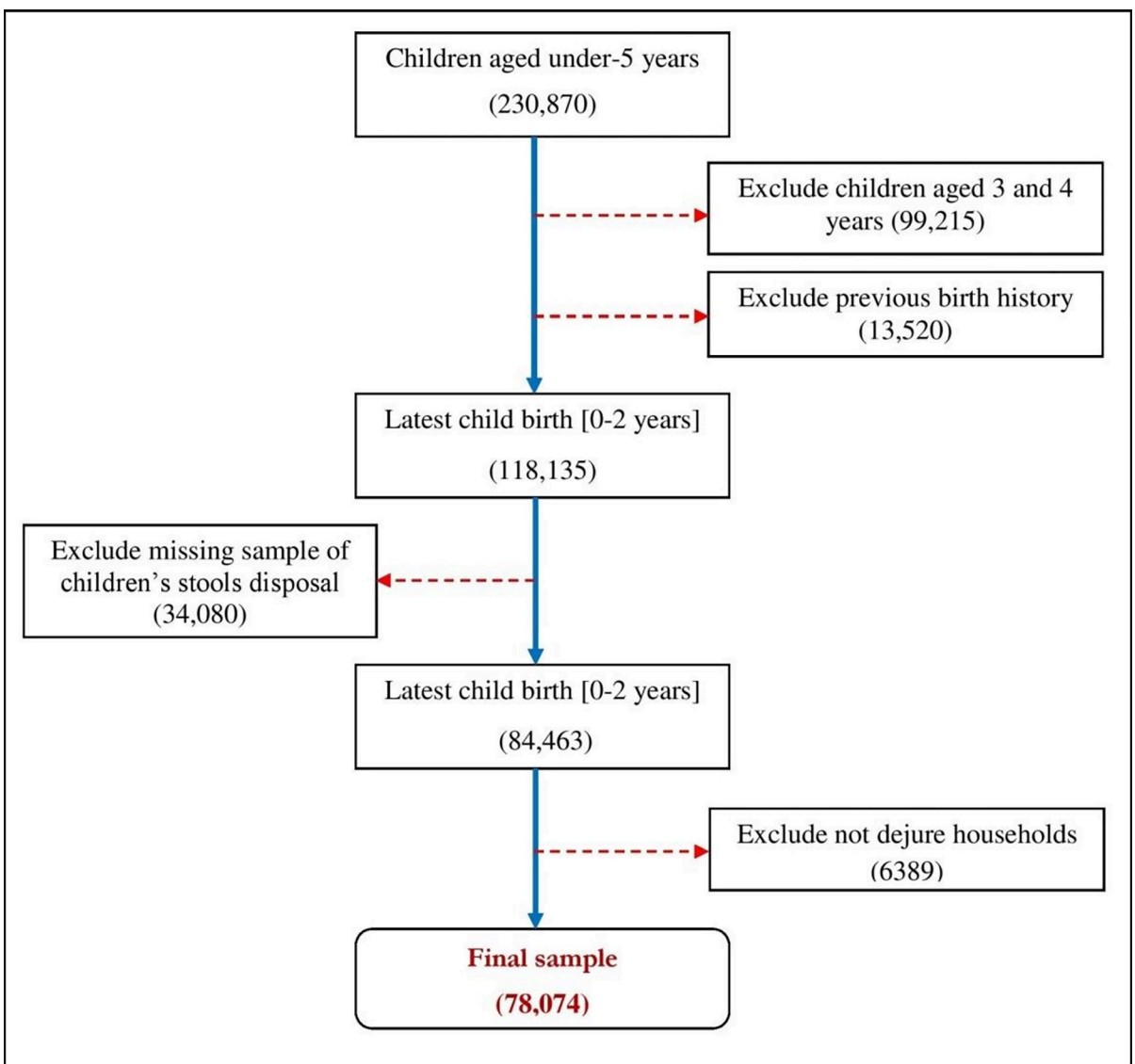

**Fig 1. Schematic diagram of the sample selection from the surveyed population.**

### Explanatory variables

The present study consisted of a set of predictors based on existing studies in different settings [11,12], including the women's age (15–19 years, 20–24 years, 25–29 years, ≥30 years), women's education (no education, primary, secondary, higher), religious affiliation (Hindu, Muslim, Christian, others), social groups (general [GEN], scheduled caste [SC], scheduled tribe [ST], other backward class [OBC]), place of residence (urban, rural), household wealth quintiles (poorest, poorer, middle, richer, richest), mass media exposure (no, partial, high), drinking water facility at premises (yes, no), sanitation facility (improved, unimproved, open defecation/no facility), geographical location (north, central, east, northeast, west, south). A detailed description of predictor variables is given in S1 Table.

## Statistical methods

Descriptive statistics were estimated in the study to describe the characteristics of the study sample. Additionally, a bivariate analysis was conducted to investigate the distribution of unsafe disposal of child stool by the selected predictors. Pearson's chi-square was used to determine the significance level of the association and degree of independence. Notably, the NFHS dataset has a hierarchical structure with households (HH), primary sampling units (PSU), and districts. Therefore, a multilevel logistic regression analysis was employed to consider the hierarchical data structure to identify potential risk factors and estimate the impact of selected analytical levels on the unsafe disposal of child stool. A four-level random intercept logistic regression model was used for the current investigation [15,19,20]. The Four-level random-intercept logistic model has been selected for the likelihood of a child under two years (*i*) in the HH *j*, PSU *k*, and district *l* being unsafe disposal of child stool ($Y_{ijkl} = 1$).

$$logit(\pi_{ijlk}) = \beta_o + BX_{ijkl} + (f_{0k} + m_{0jk} + p_{0jkl} + s_{0ijkl})$$

This model calculates the log odds of $\pi_{ijlk}$ adjusted for the vector $X_{ial}$ of predictor variables assessed at the individual level. The parameter $\beta_o$ indicates the reference category of all variables with log odds of the unsafe disposal of child stool. The random effect within the parentheses is measured as a residual differential for the district *l* ($f_{0l}$), PSU *k* ($m_{0kl}$), HH *j* ($p_{0jkl}$), and individual *i* ($s_{0ijlk}$) considered to be independent and normally distributed with mean 0 and variance $\sigma_{f0}^2$, $\sigma_{m0}^2$, $\sigma_{p0}^2$, and $\sigma_{s0}^2$, respectively. The variances were quantified between districts, PSU, and household variations. The results of multilevel logistic regression are presented in terms of adjusted odds ratios (AORs) [21,22]. All statistical analyses were performed on Stata 12 SE (Stata Corporation, College Station, Texas, USA).

## Results

### Background characteristics of the sample

Table 1 presents the background characteristics of the study sample. The majority of the mothers were between 20 and 29 years old. Nearly 20% of mothers had no formal education. A substantial number of respondents were Hindu (79%) and belonged to other backward classes (43%) in India. The percentage of respondents decreased from the bottom to upper wealth quintile. Only 6.6% of women had full exposure to the mass media, compared to almost one-third of women (28.4%) who had no exposure. Almost 30% of households had no water facility on the premises. Open defecation practice was considerable among the study population (23.4%). Around 74% of the population resided in rural areas and from the central region (28%).

### Geographical variation in prevalence of unsafe disposal of child stool

In India, the prevalence of unsafe disposal of child stool was found to be 61.3% (Fig 2). Unsafe disposal of child stool varied across the states in India, was considerably higher in Orissa (87%), Jharkhand (80%), Assam (80%) and lower in Kerala (17%), Sikkim (20%).

Subsequently, unsafe disposal of child stool also varied across the districts in India (Fig 3). It is observed that unsafe disposal of child stool is more than the national average in the eastern part of India. Moreover, it is also found to be higher in several patches of Assam, Madhya Pradesh, Andhra Pradesh and Tamil Nadu.

Table 2 presents the prevalence of unsafe disposal of child stool by socioeconomic and demographic characteristics in India. Mothers aged 15–19 (70%) and no educated mothers (75%) practiced unsafe stool disposal more than their counterparts. The prevalence of unsafe

**Table 1. Background characteristics of the study population in India, NFHS-5 (2019–21).**

| Background characteristics | N | % | 95% CI |
|---|---|---|---|
| **Mother's age (years)** | | | |
| 15–19 | 4,014 | 5.1 | 5.0–5.3 |
| 20–24 | 29,932 | 38.3 | 38.0–38.7 |
| 25–29 | 28,492 | 36.5 | 36.2–36.8 |
| ≥30 | 15,636 | 20.0 | 19.7–20.3 |
| **Mother's education** | | | |
| No education | 14,888 | 19.1 | 18.8–19.3 |
| Primary | 8,771 | 11.2 | 11.0–11.5 |
| Secondary | 40,904 | 52.4 | 52.0–52.7 |
| Higher | 13,511 | 17.3 | 17.0–17.6 |
| **Religion** | | | |
| Hindu | 61,789 | 79.1 | 78.9–79.4 |
| Muslim | 12,862 | 16.5 | 16.2–16.7 |
| Christian | 1,634 | 2.1 | 2.0–2.2 |
| Others | 1,790 | 2.3 | 2.2–2.4 |
| **Social group** | | | |
| GEN | 13,762 | 17.6 | 17.4–17.9 |
| SC | 17,950 | 23.0 | 22.7–23.3 |
| ST | 8,226 | 10.5 | 10.3–10.8 |
| OBC | 33,755 | 43.2 | 42.9–43.6 |
| Don't know | 4,382 | 5.6 | 5.5–5.8 |
| **Wealth quintile** | | | |
| Poorest | 18,578 | 23.8 | 23.5–24.1 |
| Poorer | 16,583 | 21.2 | 21.0–21.5 |
| Middle | 15,458 | 19.8 | 19.5–20.1 |
| Richer | 14,634 | 18.7 | 18.5–19.0 |
| Richest | 12,822 | 16.4 | 16.2–16.7 |
| **Mass media exposure** | | | |
| No | 22,208 | 28.4 | 28.1–28.8 |
| Partial | 50,719 | 65.0 | 64.6–65.3 |
| High | 5,147 | 6.6 | 6.4–6.8 |
| **Water facility on premises** | | | |
| Yes | 56,814 | 72.8 | 95.5–95.8 |
| No | 21,261 | 27.2 | 26.9–27.5 |
| **Sanitation facility** | | | |
| Improved | 57,580 | 74.3 | 74.0–74.6 |
| Unimproved | 1,804 | 2.3 | 2.2–2.4 |
| Open defecation | 18,130 | 23.4 | 23.1–23.7 |
| **Place of residence** | | | |
| Urban | 20,562 | 26.3 | 26.0–26.6 |
| Rural | 57,512 | 73.7 | 73.4–74.0 |
| **Region** | | | |
| North | 10,443 | 13.4 | 13.1–13.6 |
| Central | 21,648 | 27.7 | 27.4–28.0 |
| East | 20,422 | 26.2 | 25.8–26.5 |
| Northeast | 2,973 | 3.8 | 3.7–3.9 |
| West | 9,702 | 12.4 | 12.2–12.7 |

*(Continued)*

**Table 1.** (Continued)

| | | | |
|---|---|---|---|
| South | 12,886 | 16.5 | 16.2–16.8 |

Note: All samples and percentages are weighted; CI: Confidence interval.

disposal of child stool was considerably higher among Hindu followers (63.2%) and those belonging to scheduled tribes (75%) across the country. The incidence of unsafe disposal of child stool was more than two-fold higher among the poorest than the richest counterparts (82.6% vs. 34.2%) in India. The prevalence of unsafe disposal of child stool was higher among mothers who had no exposure to mass media (74.6%), households with no water facility at premises (70.6%), and unimproved sanitation facilities (70%) as compared to their counterparts, respectively. The geographical pattern showed that the rural-urban gap in the prevalence of unsafe disposal of child stool was the highest in rural (67%) and the east region (72%).

## Results from multilevel regression analyses

The results of the multilevel regression analysis are presented in Table 3, which highlights the influence of random-effect factors. In the final model, the intra-class correlation coefficient (ICC) demonstrated that household differences account for 73.1% of the overall variability in the unsafe disposal of child stool, followed by PSUs (39.9%) and districts (11.2%). The log-likelihood ratio test (LR) and logistic regression have a p-value of all <0.001 in the random effect section.

With increasing mothers' ages, the probability of unsafe disposal of child stool decreased in India. Similarly, higher educated mothers (AOR: 0.60; 95% CI: 0.53–0.69) have less likelihood of unsafe disposal of child stool than no educated mothers. Subsequently, the probability of unsafe disposal of child stool was significantly higher among scheduled tribes (AOR: 1.21; 95% CI: 1.06–1.38) than all social groups. The probability of unsafe disposal was significantly decreased from the poorest to the richest wealth quintile. Those with higher exposure to media had a lower likelihood of unsafe disposal of child stool (AOR: 0.69; 95% CI: 0.60–0.80). Other

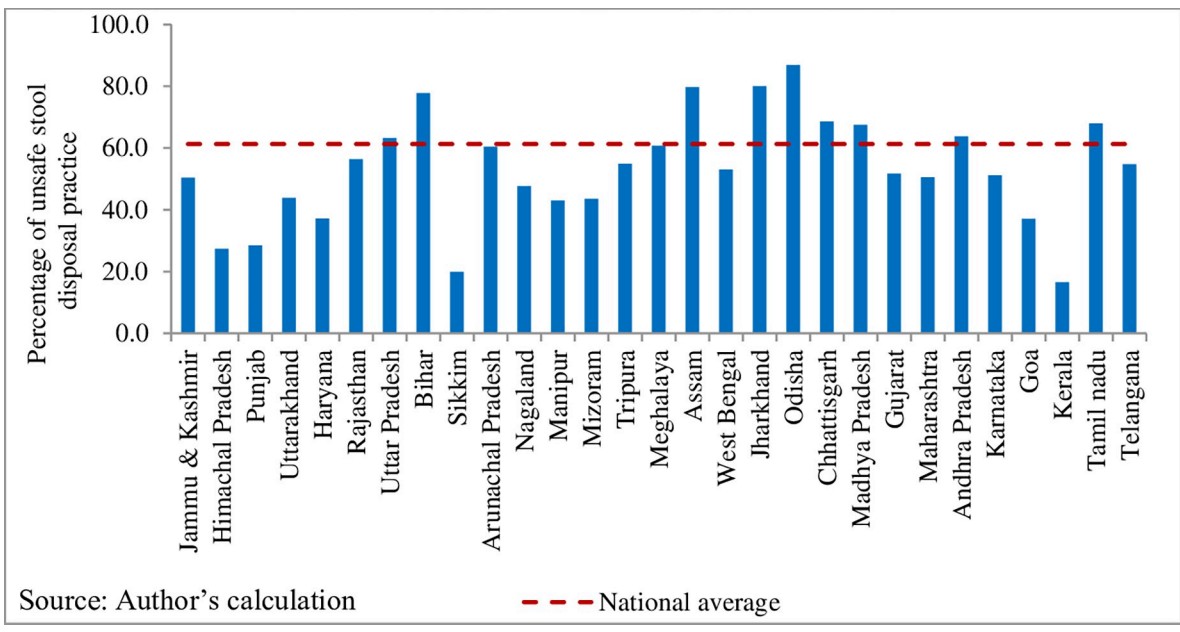

**Fig 2. State-wise variation of unsafe disposal of child stool in India, NFHS-5 (2019–21).**

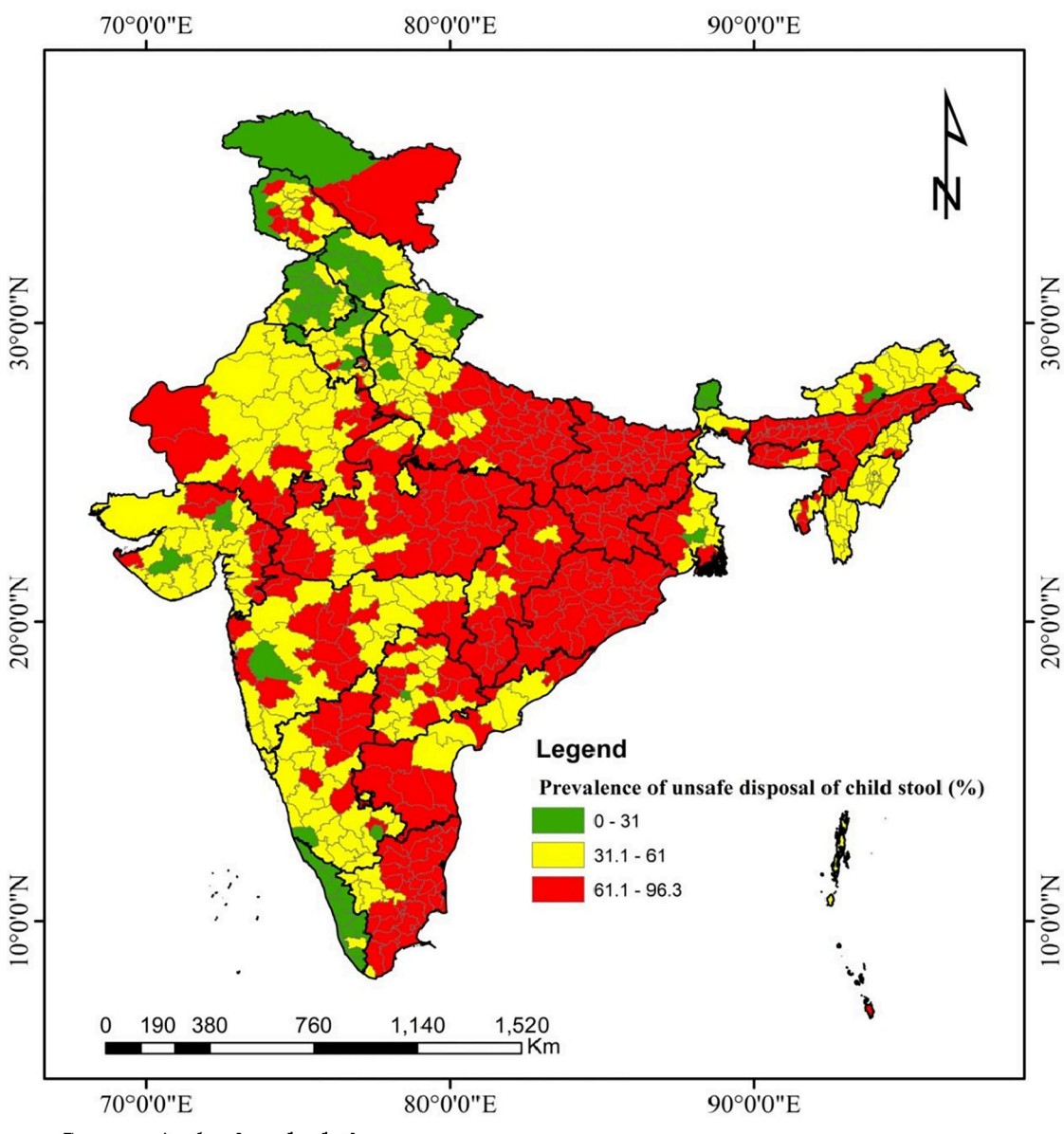

Source: Author's calculation

**Fig 3. District-wise prevalence of unsafe disposal of child stool in India, NFHS-5 (2019–21).**

sanitation-related factors such as no water facility at household premises (AOR: 1.30; 95% CI: 1.19–1.41) and unimproved sanitation facilities (AOR: 1.16; 95% CI: 0.98–1.1.37) at households had higher odds of unsafe disposal of child stool across the country. The likelihood of unsafe disposal of child stool was 20% more likely in rural areas than in urban settings. Geographical patterns demonstrated that the likelihood of unsafe disposal of child stool was significantly higher in the eastern region (AOR: 5.93; 95% CI: 4.19–8.40) than in all regions.

## Discussion

In India, significant progress has been observed in enhancing sanitation practices among adults over the past decade [10]. However, the progress and predictors of unsafe disposal of

**Table 2. Prevalence of unsafe disposal of child stool by selected background characteristics of the study population in India, NFHS-5 (2019–21).**

| Background characteristics | Unsafe stool disposal (%) | 95% CI | $\chi^2$ (*p*-value) |
|---|---|---|---|
| **Mother's age (years)** | | | |
| 15–19 | 69.9 | 68.5–71.4 | 388.1324 (<0.001) |
| 20–24 | 64.1 | 63.5–64.6 | |
| 25–29 | 59 | 58.5–59.6 | |
| ≥30 | 58 | 57.3–58.8 | |
| **Mother's education** | | | |
| No education | 75.4 | 74.7–76.1 | 2.8e+03 (<0.001) |
| Primary | 69.3 | 68.3–70.2 | |
| Secondary | 59.8 | 59.3–60.3 | |
| Higher | 45.3 | 44.4–46.1 | |
| **Religion** | | | |
| Hindu | 63.2 | 62.8–63.5 | 672.3232 (<0.001) |
| Muslim | 55.3 | 54.4–56.2 | |
| Christian | 59.1 | 56.7–61.4 | |
| Others | 43.5 | 41.2–45.8 | |
| **Social group** | | | |
| GEN | 49.4 | 48.6–50.2 | 1.3e+03 (<0.001) |
| SC | 65.5 | 64.8–66.2 | |
| ST | 75 | 74.1–76.0 | |
| OBC | 61.5 | 60.9–62.0 | |
| Don't know | 55.2 | 53.7–56.6 | |
| **Wealth quintile** | | | |
| Poorest | 82.6 | 82.0–83.1 | 8.5e+03 (<0.001) |
| Poorer | 71.4 | 70.7–72.1 | |
| Middle | 60.8 | 60.0–61.6 | |
| Richer | 47.3 | 46.5–48.1 | |
| Richest | 34.2 | 33.4–35.0 | |
| **Mass media exposure** | | | |
| No | 74.6 | 74.1–75.2 | 2.2e+03 (<0.001) |
| Partial | 57 | 56.6–57.5 | |
| High | 46.4 | 45.0–47.7 | |
| **Water facility on premises** | | | |
| Yes | 57.9 | 57.4–58.2 | 225.2120 (<0.001) |
| No | 70.6 | 70.0–71.2 | |
| **Sanitation facility** | | | |
| Improved | 53.3 | 52.9–53.7 | 5.5e+03 (<0.001) |
| Unimproved | 66.9 | 64.7–69.1 | |
| Open defecation | 86 | 85.5–86.5 | |
| **Place of residence** | | | |
| Urban | 44.7 | 44.0–45.3 | 2.4e+03 (<0.001) |
| Rural | 67.3 | 66.9–67.7 | |

(*Continued*)

**Table 2.** (Continued)

| Background characteristics | Unsafe stool disposal (%) | 95% CI | $\chi^2$ (p-value) |
|---|---|---|---|
| **Region** | | | |
| North | 47.7 | 46.7–48.6 | 3.6e+03 (<0.001) |
| Central | 64.7 | 64.0–65.3 | |
| East | 72.3 | 71.7–72.9 | |
| Northeast | 71.8 | 70.2–73.4 | |
| West | 50.9 | 49.9–51.9 | |
| South | 54.9 | 54.0–55.8 | |

Note-All percentages are weighted; CI: Confidence interval.

child stool have been limited in the Indian context. Despite the evidence highlighting the negative impact of the unsafe disposal of child stool on child health in India [3], nuanced socio-cultural and geographical factors remain unexplored using the latest nationally representative dataset [1,17,18]. This study addresses this research gap, comprehensively examining the unsafe disposal of child stool among mothers of children under two years. In particular, the study investigates the level, patterns, and determinants of unsafe disposal of child stool in India by highlighting geographical unit-level variation.

The present study found that a majority of mothers with children under two years practice unsafe disposal of child stool (61%) in India. This prevalence remains lower than Angola (68%), Benin (66%), and rural Bangladesh (81%) [15,23,24] but it is higher than in Burundi (33%), Cameroon (27%), Malawi (15%), Mali (36%), Rwanda (14%), Uganda (20%), Zambia (22%), and Zimbabwe (14%) [15]. A significant variation in unsafe disposal of child stool exists at different geographical levels in India, such as state level (Fig 2), district level (Fig 3), community level (PSU), and household level (Table 3). The variations observed in different sampling units can be attributed to the diverse socioeconomic, cultural, and household-level infrastructure influencing unsafe disposal practices in each unit. The findings from the multilevel logistic regression model show that the predictors such as mother's age, education, mass media exposure, religion, social group, wealth quintiles, water connectivity, and improved sanitation facility at household, residence, and region are significantly associated with the unsafe disposal of child stool. These results align with micro-level studies in India [1,17,18] and large-scale studies conducted in Bangladesh and sub-Saharan Africa [11–13,15,23].

Unsafe disposal of child stool was significantly linked to the mother's age. Among the mothers aged 15–19 years, the likelihood of unsafe disposal of child stool is higher than in the higher age groups. Mothers with early marriage and early childbearing, particularly teenagers, have lower levels of education and limited resources in the household. These perhaps resulted in lower awareness of safe disposal practices of child stool and a lack of resources to manage the safe disposal practices [10]. Interestingly, this trend aligns with Sub-Saharan Africa [15] but contradicts Nigerian [24] and Gambian studies [25], likely due to socio-cultural and socio-demographic variations across countries, which significantly shape stool disposal dynamics and observed differences [25].

Aligning with prior studies [15,18], the current study found that maternal education and mass media exposure are negatively associated with the unsafe disposal of child stool in India. The prior study suggests that mothers with a higher level of education and a greater level of mass media exposure are more aware of the associated risks of unsafe disposal of child stool, adopting safer practices and healthier lifestyles [15,18]. Moreover, the mothers who had access to the media may have heard essential health information regarding the proper disposal of

**Table 3. Adjusted odds ratio (AOR) and random intercept statistics of unsafe disposal of child stool by background characteristics of the study population in India, NFHS-5 (2019–21).**

| Background characteristics | AOR | 95% CI |
|---|---|---|
| **Mother's age (years)** | | |
| 15–19Ⓡ | 1.00 | |
| 20–24 | 0.79** | 0.67–0.92 |
| 25–29 | 0.67*** | 0.57–0.78 |
| ≥30 | 0.63*** | 0.54–0.74 |
| **Mother's education** | | |
| No educationⓇ | 1.00 | |
| Primary | 0.97 | 0.86–1.09 |
| Secondary | 0.77*** | 0.69–0.85 |
| Higher | 0.60*** | 0.53–0.69 |
| **Religion** | | |
| HinduⓇ | 1.00 | |
| Muslim | 0.80*** | 0.71–0.90 |
| Christian | 0.82 | 0.65–1.02 |
| Others | 0.72** | 0.59–0.89 |
| **Social group** | | |
| GENⓇ | 1.00 | |
| SC | 1.13** | 1.03–1.17 |
| ST | 1.21** | 1.06–1.38 |
| OBC | 0.96 | 0.87–1.05 |
| Don't know | 0.86 | 0.72–1.02 |
| **Wealth quintile** | | |
| PoorestⓇ | 1.00 | |
| Poorer | 0.64*** | 0.58–0.71 |
| Middle | 0.41*** | 0.36–0.47 |
| Richer | 0.23*** | 0.20–0.27 |
| Richest | 0.13*** | 0.11–0.16 |
| **Mass media exposure** | | |
| NoⓇ | 1.00 | |
| Partial | 0.92* | 0.84–1.00 |
| High | 0.69*** | 0.60–0.80 |
| **Water facility on premises** | | |
| YesⓇ | 1.00 | |
| No | 1.30*** | 1.19–1.41 |
| **Sanitation facility** | | |
| ImprovedⓇ | 1.00 | |
| Unimproved | 1.16* | 0.98–1.37 |
| Open defecation | 4.74*** | 4.10–5.48 |
| **Place of residence** | | |
| UrbanⓇ | 1.00 | |
| Rural | 1.20*** | 1.15–1.26 |
| **Region** | | |
| NorthⓇ | 1.00 | |
| Central | 2.92*** | 2.14–3.97 |
| East | 5.93*** | 4.19–8.40 |
| Northeast | 2.20*** | 1.55–3.12 |

*(Continued)*

**Table 3.** (Continued)

| Background characteristics | AOR | 95% CI |
|---|---|---|
| West | 1.29 | 0.89–1.86 |
| South | 2.06*** | 1.50–2.84 |
| **Constant** | 4.23*** | 3.11–5.77 |
| **Random intercept parameter** | | |
| Var (district) | 1.36 | 1.13–1.65 |
| Var (PSU) | 3.52 | 3.03–4.08 |
| Var (HHs) | 4.06 | 3.20–5.14 |
| ICC (district) (%) | 11.2 | |
| ICC (PSU) (%) | 39.9 | |
| ICC (HHs) (%) | 73.1 | |
| **Model fit statistics** | | |
| Wald test $\chi^2$ | 699.57*** | |
| LR test vs. logistic regression | <0.001 | |

Note- Ⓡ = Reference category, Significance level at

***≤0.001

**≤0.01

*≤0.05

AOR = Adjusted odds ratio, CI = Confidence interval.

child waste and its effects on the child's health and the community. As a result, these mothers may have developed a positive attitude towards the significance of safe disposal practices of child stool and a better understanding of safe child waste disposal.

Consistent with earlier studies in India and elsewhere [1,17,18,26], the findings from this study affirm that improving household wealth parallels a decline in unsafe disposal of child stool. A higher household wealth facilitates mothers' better living conditions, including upgraded sanitation facilities. This transformation cultivates healthier habits, curbs unsafe disposal practices, and nurtures hygiene practices [26]. Similar to previous studies [1,25,26], water and improved sanitation facilities appeared as pivotal household factors influencing disposal practices of child stool. The current study explored that households lacking water facilities, especially in rural India, are prone to unsafe disposal. Similarly, unimproved sanitation and open defecation are positively linked with unsafe disposal of child stool. Prior research highlighted the significant role of water connectivity and improved sanitation facilities at the house for adult hygiene in India [10]; the current study also expands its significance for the safe disposal of child stool.

The rural-urban divide in unsafe disposal of child stool exists in India. To bridge this gap between rural and urban areas, the current study highlights that there is a need to refocus on ongoing interventions like Swachh Bharat Abhiyan for improved sanitation, Saakshar Bharat Abhiyan for female literacy promotion, and Jal Jeevan Mission for expansion of piped water access in rural India. The current study also recommends further study to examine the factors contributing to the rural-urban divide in unsafe disposal of child stool in India.

The current study findings suggest that micro-level variance in the unsafe disposal of children's stool was higher than at the macro level (Table 3). In the overall variance, HH explained 73% of the variance in unsafe disposal, followed by PSU at 40% in India. Given the substantial influence of households on the overall variance, programmatic interventions at the household level need to be prioritized. While household-level factors played a crucial role in the

geographical variance of unsafe disposal of children's stool, community-level factors also significantly contributed to this. Therefore, the findings underscore the importance of community engagement in promoting the safe disposal of child stool.

### Strengths

This study boasts several noteworthy strengths. Foremost, it is a pioneering effort to contextualize the unsafe disposal of child stool in India through a sophisticated multilevel analysis. By delving into different geographical levels, the study provides a comprehensive grasp of the prevalence and predictors of unsafe disposal practices in India. Furthermore, the study leverages the extensive reach of the latest NFHS data (2019–21), both in characteristics of a sizable sample and national representative nature. Rigorous statistical techniques, including sample weighting to counterbalance non-proportional sample allocation, contribute to bolstering the study's statistical robustness, thereby enhancing its validity.

### Limitations

Notwithstanding its strengths, this study has certain limitations that warrant consideration. Being a cross-sectional study, establishing a causal relationship between outcome and independent variables remains challenging. Moreover, certain pertinent variables, such as the level of knowledge concerning the disposal practices of child stool, were absent from the NFHS dataset, a potentially confounding factor of the analysis. The reliance on self-reported data introduces the possibility of both social desirability bias and recall bias, compromising the absolute accuracy of the findings. It is important to note that the study's depiction of prevalence and predictors is anchored in the present moment, inadvertently neglecting the potential evolution of practices over time. Lastly, owing to constraints imposed by data availability and the inherent nature of cross-sectional studies, specific qualitative nuances such as cultural norms, habits, and beliefs [27] that could intricately influence results were not captured.

### Conclusion

In summary, this comprehensive study casts a spotlight on the persisting issue of unsafe disposal of child stool among mothers with children under two years old in India. Despite notable strides in public health development, a crucial gap remains in addressing this vital aspect of child hygiene. The investigation unravels an intricate interplay of socio-cultural, geographical, and household factors that shape these unsafe disposal practices. Maternal education, exposure to mass media, household prosperity, water accessibility, and improved sanitation facilities emerge as pivotal determinants. Moreover, the rural-urban divide in predictors of unsafe disposal of child stool is considerable in India. HH-level variance in the unsafe disposal of children's stool was significant. Therefore, the findings emphasize the need for targeted interventions, such as target-based poverty alleviation programs, improved sanitation and water facilities initiatives, and community-level public health awareness programs. To ensure the holistic well-being of young children, these insights call for a concerted effort to bridge existing gaps and enhance child hygiene practices across diverse contexts in India.

### Supporting information

**S1 Table. Operational description of the predictor variables.**
(DOCX)

## Author Contributions

**Conceptualization:** Margubur Rahaman, Avijit Roy, Md. Juel Rana.

**Data curation:** Margubur Rahaman, Avijit Roy.

**Formal analysis:** Margubur Rahaman, Pradip Chouhan, Md. Juel Rana.

**Investigation:** Avijit Roy, Pradip Chouhan, Md. Juel Rana.

**Methodology:** Margubur Rahaman, Avijit Roy, Pradip Chouhan, Md. Juel Rana.

**Resources:** Avijit Roy.

**Software:** Margubur Rahaman, Avijit Roy, Md. Juel Rana.

**Supervision:** Avijit Roy, Md. Juel Rana.

**Validation:** Margubur Rahaman, Avijit Roy, Pradip Chouhan.

**Visualization:** Avijit Roy, Pradip Chouhan.

**Writing – original draft:** Margubur Rahaman, Avijit Roy, Md. Juel Rana.

**Writing – review & editing:** Margubur Rahaman, Avijit Roy, Pradip Chouhan, Md. Juel Rana.

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
