## [Decision Letter · Decision Letter 0]

8 Nov 2023

PONE-D-23-28796Exploring drivers of unsafe young children's stool disposal practices in India using hierarchical regression model and Fairlie decomposition analysisPLOS ONE

Dear Dr. Rahaman,

Thank you for submitting your manuscript to PLOS ONE. After careful consideration, we feel that it has merit but does not fully meet PLOS ONE’s publication criteria as it currently stands. Therefore, we invite you to submit a revised version of the manuscript that addresses the points raised during the review process.

We look forward to receiving your revised manuscript.

Kind regards,

Pintu Paul

Academic Editor

PLOS ONE

Reviewers' comments:

Reviewer's Responses to Questions

**Comments to the Author**

1. Is the manuscript technically sound, and do the data support the conclusions?

Reviewer #1: Yes

Reviewer #2: Yes

2. Has the statistical analysis been performed appropriately and rigorously? 

Reviewer #1: Yes

Reviewer #2: Yes

3. Have the authors made all data underlying the findings in their manuscript fully available?

Reviewer #1: Yes

Reviewer #2: Yes

4. Is the manuscript presented in an intelligible fashion and written in standard English?

Reviewer #1: Yes

Reviewer #2: Yes

5. Review Comments to the Author

Reviewer #1: I read with interest the manuscript in which they exploring drivers of unsafe young children's stool disposal practices in India using hierarchical regression model and Fairlie decomposition analysis.

Minor comments:

• The introduction is far too long.

• In the Introduction section, author/authors are written “proper stool management is crucial to decrease short- and long-term health risks among children under five”. But why only includes recent birth children between the ages of under 2 years?

• Write more about water facility at premises in the Explanatory variables section.

• Please provide the more details of Model validation.

• Write more about why author excluded not dejure households from study population.

• The data interpretation is well presented, the statistical significance of obtained data is little needed to improve.

• In heading of the column in Table 3 author have written Rural, Urban, again, Urban. I think, this will be India.

• The comparison of data obtained in the present study to published data should be added along with a clear description of what new the present study adds to the current knowledge, a more detailed exposition of the points for a deeper understanding would be desirable.

Reviewer #2: Thank you for providing me the opportunity to review this paper. The authors' manuscript is well-written, and the statistical application is also good. There were some minor issues regarding the writing; therefore, I recommend it for publication with minor revisions. The suggestions are as follows:

1. In abstract result section please replace wording “primarily attributed” to “contributing significant were”. Similarly, please replace “as influential in this practice” to “were significant predictors” for readers’ clarity.

2. In introduction section, please replace “conspicuous divide” to “regional divide”.

3. Please add predictors of unsafe young children’s stool disposal practices in the sentence – “Therefore, a study is relevant to contextualize the disposal practice of child stools in India”.

4. I suggest authors to rewrite this sentence and replace “children utlize” to “mother utilize”- This encompasses situations where children utilize designated toilets or latrines for stool disposal.

5. As the research on children stool management, so better to use mother’s age, education throughout the manuscript instead of women age or education and others to mother’s age, education.

6. Please add abbreviation of CI below the table 1.

7. I suggest authors to add a figure- prevalence of unsafe stool disposal by place of residence and national average in India.

8. In table 5 please maintain only point two-digit reporting.

9. However, a gap persists in understanding unsafe stool disposal practices among young children. - Please add “a research gap” instead of” gap” in this sentence.

10. Authors used the wording “household wealth”, please write wealth quintile.

11. Major findings-based policy suggestions missing in the discussion section. I suggest authors to add your major finding-based policy implications.

12. Please rewrite this sentence- Utilizing secondary data from the fifth round of the National Family Health Survey (2019- 21), the study examined 78,074 births under two years.

13. I found some minor grammatical errors and inconsistency of wording, so please correct it carefully.

14. The title says “Exploring drivers of unsafe young children's stool disposal practices in India using hierarchical regression model and Fairlie decomposition analysis”. From the title it is not clear which gap you want to decompose using Fairlie decomposition.

15. In abstract it is written “The study aimed to identify the factors behind this unsafe practice in India.” But the study also decomposes the rural urban gap in child stool disposal. Please mention it accordingly.

16. Abstract: “Fairlie decomposition indicated a 21% rural-urban disparity”. Fairlie decomposition decomposes the gap or inequality; it does not 'indicate' the inequality. So please rectify the sentence. Please mention how much rural-urban gap is explained or decomposed using Fairlie decomposition.

17. Please shorten the section “Sampling technique”. Citing the NFHS-5 report would also do the job. No need to discuss NFHS-5 sampling technique in such details.

18. Authors mentioned “The chi-square statistic was used to assess the significance of the selected predictors and outcome variables”. However I couldn’t find out the Chi-squared results.

6. PLOS authors have the option to publish the peer review history of their article (what does this mean?). If published, this will include your full peer review and any attached files.

Reviewer #1: **Yes: **Jay Saha

Reviewer #2: No

---

## [Author Response · Author response to Decision Letter 0]

26 Nov 2023

Dear Editor,

We are resubmitting our revised paper to Plos One titled “Exploring drivers of unsafe disposal of child stool in India using hierarchical regression model” the original title is “Exploring drivers of unsafe young children's stool disposal practices in India using hierarchical regression model and Fairlie decomposition analysis”. We would like to express our sincere gratitude for your careful review of our manuscript. We have made revisions to our paper again after paying close attention to the comments and suggestions by the reviewers. Your insightful queries and valuable comments have greatly contributed to improving the scientific rigor and clarity of our study, and have helped us to enhance the quality of our manuscript.

Please find our manuscript enclosed, which has been thoroughly revised in response to your comments and suggestions. Below is a detailed point-by-point response to the reviewers’ comments.

Reviewer #1: 

I read with interest the manuscript in which they exploring drivers of unsafe young children's stool disposal practices in India using hierarchical regression model and Fairlie decomposition analysis. 

Minor comments:

• The introduction is far too long.

Response: Thank you for your observation. We tried to capture comprehensive scenario of unsafe child stool practice globally. Therefore, the background of the study is long as you mentioned. We hope it will help to the readers for clear understanding.

• In the Introduction section, author/authors are written “proper stool management is crucial to decrease short- and long-term health risks among children under five”. But why only includes recent birth children between the ages of under 2 years?

Response: Thank you for your query. Please see the revised manuscript (page: 6; line 140-143).

• Write more about water facility at premises in the Explanatory variables section.

Response: Thank you for your query. Water facility is classified into two categories as per demographic health survey (DHS) guideline. It is difficult to describe all things in manuscript. You can visit NFHS-5 report for better understanding. Please look at the website for your query: 

https://rchiips.org/nfhs/NFHS-5Reports/National%20Report%20Volume%20II.pdf

• Please provide the more details of Model validation.

Response: Thank your sir for your suggestion. We think the manuscript has sufficient description about all models. We have also checked VIF also. 

• Write more about why author excluded not dejure households from study population.

Response: We exclude not dejure households for avoiding biasness in study sample.

• The data interpretation is well presented, the statistical significance of obtained data is little needed to improve.

Response: Thank your sir for your suggestion. We have incorporated all things as per your suggestion.

• In heading of the column in Table 3 author have written Rural, Urban, again, Urban. I think, this will be India.

Response: We really thankful to you. Yes, you’re absolutely correct. We rectified as India. Thank you again.

• The comparison of data obtained in the present study to published data should be added along with a clear description of what new the present study adds to the current knowledge, a more detailed exposition of the points for a deeper understanding would be desirable.

Response: Thank your sir for your suggestion. We addressed your suggestion thoroughly in our revised manuscript. Please see the introduction and discussion section.

Reviewer #2: 

Thank you for providing me the opportunity to review this paper. The authors' manuscript is well-written, and the statistical application is also good. There were some minor issues regarding the writing; therefore, I recommend it for publication with minor revisions. The suggestions are as follows:

1. In abstract result section please replace wording “primarily attributed” to “contributing significant were”. Similarly, please replace “as influential in this practice” to “were significant predictors” for readers’ clarity.

Response: Thank you sir for your nuance observation. We replaced the words as per your recommendations. Please see the revised manuscript (line 41 and 42). 

2. In introduction section, please replace “conspicuous divide” to “regional divide”.

Response: We replaced the words as ‘regional divide’ as you mentioned. Please see the revised manuscript. 

3. Please add predictors of unsafe young children’s stool disposal practices in the sentence – “Therefore, a study is relevant to contextualize the disposal practice of child stools in India”.

Response: Thank you for your observation. We replaced the words as per your recommendations. Please see the revised manuscript. 

4. I suggest authors to rewrite this sentence and replace “children utlize” to “mother utilize”- This encompasses situations where children utilize designated toilets or latrines for stool disposal.

Response: We replaced the word as per your recommendations. Please see the revised manuscript (line 163). 

5. As the research on children stool management, so better to use mother’s age, education throughout the manuscript instead of women age or education and others to mother’s age, education.

Response: We replaced it women’s to mother’s age/education and vice-versa throughout the revised manuscript. 

6. Please add abbreviation of CI below the table 1.

Response: We incorporated abbreviation of CI in Tables

7. I suggest authors to add a figure- prevalence of unsafe stool disposal by place of residence and national average in India.

Response: Thank you for your suggestion. Please see the Figure 2. We also included district wise prevalence map

8. In table 5 please maintain only point two-digit reporting.

Response: Thank you for your observation. We exclude decomposition analysis.

9. However, a gap persists in understanding unsafe stool disposal practices among young children. - Please add “a research gap” instead of” gap” in this sentence.

Response: Thank you for your observation. We add this in our revised manuscript (line 274).

10. Authors used the wording “household wealth”, please write wealth quintile.

Response: We used ‘wealth quintile’ instead of ‘household wealth quintile’ as per your suggestion in all necessary places in our revised manuscript.

11. Major findings-based policy suggestions missing in the discussion section. I suggest authors to add your major finding-based policy implications.

Response: Thank you for your recommendation. Please see the discussion section (line 295-298 and 307-310; 323-328)

12. Please rewrite this sentence- Utilizing secondary data from the fifth round of the National Family Health Survey (2019- 21), the study examined 78,074 births under two years.

Response: We revised the sentence as “The study used 78,074 births under two years from the fifth round of the National Family Health Survey (2019-21)” (line 34-37) .

13. I found some minor grammatical errors and inconsistency of wording, so please correct it carefully.

Response: We rectified all necessary grammatical errors in our revised manuscript.

14. The title says “Exploring drivers of unsafe young children's stool disposal practices in India using hierarchical regression model and Fairlie decomposition analysis”. From the title it is not clear which gap you want to decompose using Fairlie decomposition.

Response: We revised our study title as “Exploring drivers of unsafe disposal of child stool in India using hierarchical regression model” 

15. In abstract it is written “The study aimed to identify the factors behind this unsafe practice in India.” But the study also decomposes the rural urban gap in child stool disposal. Please mention it accordingly.

Response: Thank you for your observation. We replaced as “the current study aims to identify the socioeconomic and demographic factors associated with the unsafe disposal of child stool in India and to estimate the geographical variation in unsafe disposal.

16. Abstract: “Fairlie decomposition indicated a 21% rural-urban disparity”. Fairlie decomposition decomposes the gap or inequality; it does not 'indicate' the inequality. So please rectify the sentence. Please mention how much rural-urban gap is explained or decomposed using Fairlie decomposition.

Response: We exclude decomposition analysis.

17. Please shorten the section “Sampling technique”. Citing the NFHS-5 report would also do the job. No need to discuss NFHS-5 sampling technique in such details.

Response: Thank you for your observation. We removed these sections ‘Study design and setting’ and ‘Sampling technique’ in our revised manuscript.

18. Authors mentioned “The chi-square statistic was used to assess the significance of the selected predictors and outcome variables”. However I couldn’t find out the Chi-squared results.

Response: Thank you for your observation. We added chi-squared value in table 2.

---

## [Editor Report · Decision Letter 1]

29 Nov 2023

Exploring drivers of unsafe disposal of child stool in India using hierarchical regression model

PONE-D-23-28796R1

Dear Dr. Rahaman,

We’re pleased to inform you that your manuscript has been judged scientifically suitable for publication and will be formally accepted for publication once it meets all outstanding technical requirements.

Kind regards,

Pintu Paul

Academic Editor

PLOS ONE
---

## [Editor Report · Acceptance letter]

8 Mar 2024

PONE-D-23-28796R1 

PLOS ONE

Dear Dr. Rahaman, 

I'm pleased to inform you that your manuscript has been deemed suitable for publication in PLOS ONE. Congratulations! Your manuscript is now being handed over to our production team.

Kind regards, 

on behalf of

Dr. Pintu Paul 

Academic Editor

PLOS ONE